# Heart Rate Lowering Significantly Increases Feasibility in Doppler Recording Blood Flow Velocity in Coronaries during Transthoracic Doppler Echocardiography

**DOI:** 10.3390/diagnostics13040670

**Published:** 2023-02-10

**Authors:** Carlo Caiati, Paolo Pollice, Mario Erminio Lepera

**Affiliations:** Institute of Cardiovascular Disease, Department of Interdisciplinary Medicine, University of Bari “Aldo Moro”, 70124 Bari, Italy

**Keywords:** coronary blood flow Doppler recording, coronary flow reserve, left circumflex blood flow Doppler recording, enhanced transthoracic Doppler, left anterior descending coronary artery, lowering heart rate

## Abstract

**Background**: Coronary blood flow Doppler recording by Transthoracic Doppler in convergent mode (E-Doppler TTE) might be further improved by lowering heart rate (HRL) down to <60 bpm, since low HR < 60 b/m causes a disproportional lengthening of the diastole, so the coronaries are still for a longer time, very much improving the Doppler signal/noise ratio. **Methods**: A group of 26 patients underwent E-Doppler TTE before and after HR lowering in four branches of the coronary tree, namely, the left main (LMCA); left anterior descending (LAD), subdivided into three segments: proximal, mid and distal; proximal left circumflex (LCx); and obtuse marginal (OM). Color and PW coronary Doppler signal was judged by two expert observers as undetectable (SCORE 1), weak or with clutter artifacts (SCORE 2), or well delineated (SCORE 3). In addition, local accelerated stenotic flow (AsF) was measured in the LAD before and after HRL. **Results**: Beta-blockers significantly decreased the mean HR from 76 ± 5 to 57 ± 6 bpm (*p* < 0.001). Before HRL, the Doppler quality was very poor in the proximal and mid-LAD segments (median score value = 1 in both), while in the distal LAD, it was significantly better but still suboptimal (median score value = 1.5, *p* = 0.009 vs. proximal and mid-LAD score). After HRL, blood flow Doppler recording in the three LAD segments was strikingly improved (median score value = 3, 3 and 3, *p* = ns), so the effect of HRL was more efficacious in the two more proximal LAD segments. In 10 patients undergoing coronary angiography (CA), no AsF as expression of transtenotic velocity was detected at baseline. After HRL, thanks to the better quality and length of color flow, ASF was detected in five patients while in five others, it was not in perfect agreement with CA (Spearman correlation coefficient = 1, *p* < 0.01). The color flow in the proximal LCx and OM was extremely poor at baseline (color flow length 0 and 0, median (interquartile range) mm, respectively) and improved considerably after HRL (color flow length 23 [13.5] and 25 [12.0] mm, respectively, *p* < 0.001). **Conclusions**: HRL greatly improved the success rate of blood flow Doppler recording in coronaries, not only in the LAD, but also in the LCx. Therefore, AsF for stenosis detection and coronary flow reserve assessment can have wider clinical applications. However, further studies with larger samples are needed to confirm these results.

## 1. Introduction

Coronary blood flow Doppler recording by enhanced transthoracic Doppler echocardiography (E-Doppler TTE) is a very important approach for coronary artery disease (CAD). The two main clinical applications are coronary flow reserve assessment [1] and the detection of an acceleration of blood flow at the stenosis site [2,3,4,5]. However, it can be a challenging task, even if, recently, different strategies (new beam former transducer with higher frequency, contrast and second harmonic technologies, new tomographic plane orientation) have enhanced the feasibility of this approach to some extent [1,6,7,8,9]. Notwithstanding the attempts to improve the method, it has never gained a wide clinical application because the approach remains challenging and has suboptimal feasibility.

Heart rate lowering (HRL) below 60 b/min can disproportionally increase the duration of the diastolic time [10]. If the diastolic time is prolonged, coronary insonification is potentially more successfully obtained, since, during the diastole, the coronaries are still and coronary blood flow signal is not cluttered by tissue movement. A recent report has demonstrated that HR lowering can greatly improve the feasibility of BF (blood flow) Doppler recording in the left anterior descending coronary artery (LAD) and left main coronary artery (LMCA) [3]. However, it is not known if the enhancing effect uniformly affects the whole LAD or creates more benefit for only some segments of the LAD. In fact, the cluttering effect of tissue movement may be more disturbing for those segments of LAD, such as the proximal and mid-LAD segments, closer to the base of the heart, that is the more mobile part [11,12]. Contrarily, visualization of the distal LAD segments that run in the anterior distal groove, part of the apex of the heart (scarcely mobile part of the heart), may be less benefited by the low HR [12].

Moreover, it is not known if, in the same set of patients in which the quality Doppler signal improvement of coronary blood flow has been documented, HRL in the LAD can really drive more meaningful clinical information regarding coronary atherosclerosis in this artery. In fact, coronary atherosclerosis narrowing, both severe or subcritical, can be detected very reliably and directly by E-Doppler TTE, as recently demonstrated, if LAD blood flow is appropriately Doppler-explored [3,4,13,14]. Coronary atherosclerosis segmental narrowing creates an acceleration of BF velocity at the stenosis site (AsF), that is reliably recorded by E-Doppler TTE if the quality of the Doppler signal is good enough and the stenotic segment is appropriately insonified. Since most plaques are located in the proximal and mid-LAD [15], it is essential that this part of the LAD is insonified as thoroughly as possible.

Finally, no data have been reported on the effect of HR reduction on the feasibility of BF Doppler recording in the Circumflex Coronary artery (LCx); BF Doppler recording in LCx can be hard to be obtain, essentially due to the deep location in the chest of this branch, the very large mass of lung tissue covering this artery and also the anatomic complexity and variation of this posterior coronary branch [16]. All these anatomic aspects are compounded, like in the LAD, by the elevated mobility and velocity of movements of these arteries, since their course is close to the base of the heart, which moves very rapidly during contraction, hampering Doppler recording [12].

Therefore, we first hypothesized that HR lowering below 60 b/m is more beneficial for the proximal LAD segments than the distal; second, that HR lowering improves feasibility in BF Doppler recording in the proximal–mid circumflex coronary artery and in the obtuse marginal branch (OM); and finally, that this better quality of BF Doppler recording after HR reduction in the LAD should bring about an appropriate detection of coronary stenoses. For this purpose, 26 patients with HR > 60 b/m scheduled for E-Doppler TTE and, if necessary, coronary angiography, were enrolled.

## 2. Materials and Methods

A group of 26 unselected patients with suspected CAD and baseline HR > 65 b/m underwent E-Doppler-TTE between July and December 2017, before and after HR lowering, in 4 branches of the coronary tree: left main coronary artery (LMCA), LAD, proximal LCx and MO.

### 2.1. Echocardiographic Equipment Characteristics and Settings 

Echocardiography was performed using an Acuson Sequoia™ ultrasound unit (C256 Echocardiography System, Siemens Healthcare, Erlangen, Germany) and broadband transducer (3V2c). The color Doppler signal was attained in convergent color Doppler mode at 2.5 or 2.0 MHz transmission frequency, while the spectral Doppler was performed in the fundamental mode at 2.5 or 2.0 MHz. The color-coded Doppler setting was adjusted to maximize scanning sensitivity (pulse repetition frequency was set at 16 cm/s [2.5 MHz] or 20 cm/s [2.0 MHz] with minor modulation in special cases, and maximizing the sample volume of color flow mapping scan lines) without significantly reducing the frame rate (the color box size was reduced to remain in keeping with a frame rate of >30 Hz). All the studies were digitally stored on the built-in dedicated hard drive.

### 2.2. Echo Tomographic Planes

As previously validated, in order to visualize coronary blood flow in LMCA and LAD, we used standard parasternal and the new both parasternal and apical approaches [3,4]. The proximal LAD goes from the LMCA bifurcation down to the pulmonary annulus (internal point of reference for its identification is the pulmonary valve); the mid goes from the pulmonary annulus caudally as far as possible in the interventricular groove; the distal is that segment visualized in the distal interventricular groove with a modified 2-chamber view [17].

Briefly, after obtaining the short axis view of the aorta from the parasternal view, the left coronary fossa was identified as the echo-dense region adjacent to the left coronary sinus, delimited by the left pulmonary artery above and at the summit of the left ventricle below. Then, we attempted to visualize the LAD in this area by slightly angling the transducer up and down, and gradually rotating it clockwise in order to deal with the variable inclination of the vessel in the vertical plane (a 0–90° angle) (Figure 1). Once identified in the B-mode and then with colour Doppler, the course of the LAD was followed as far as possible.

The mid-LAD was visualized using a lower parasternal short axis view of the base of the heart, modified by a slight clockwise rotation of the transducer beam, which allows the transection of the upper interventricular portion of the artery running laterally to the right ventricular outflow tract (RVOT) to be seen before it becomes completely vertical [4].

Visualization of the proximal and mid-LAD was recently improved by moving the transducer as far as possible to the left, exploiting the variable extension of the cardiac notch of the left lung in order to better transect the content of the interventricular sulcus, since this approach can better tackle the prominence of the right ventricle outflow tract and better transect the depressed sulcus content where the artery runs. At the same time, some portion of the proximal LAD is insonified by pointing the transducer toward the aortic root [3].

The distal LAD was visualized using a modified two-chamber view with a slight medial angling of the transducer, as previously reported [1].

The apical approach for LAD, rarely used in this series of patients, is described elsewhere [3].

The insonification of proximal LCx was performed using a similar approach to that for the proximal LAD. However, LCx runs were directed posteriorly, with an almost vertical course along the left side of the ostium of the left ventricle down to the inferior LV wall (so diverging from the orientation of the proximal LAD, that is much more horizontal and directed anteriorly). Therefore, to insonify this artery with this orientation from the anterior chest wall, the only successful approach is to scan using an extremely high parasternal approach and directing the plane downward [18]. With this approach, inevitably only a minimal part of the LAD is transected, since these two arteries run in different planes. The internal marker of reference that aids in identifying the artery and the atrioventricular sulcus is the left atrial appendage.

The insonification of the obtuse marginal branch consists of transecting from the apex the obtuse margin of the left ventricle, which can be defined as the border between the postero-inferior and lateral wall of the left ventricle [18]. To do that from an apical four-chamber view, first it is best to try to visualize as well as possible the left lateral wall at full thickness (from the endocardium to the epicardium). To reach this aim, even a foreshortened 4-chamber view can be tried in order to improve quality. Then, the lateral wall should be moved as much as possible toward the center of the sector, where the maximal intensity of the ultrasound energy is present. Then, a 5-chamber view has to be obtained in order to transect the obtuse margin of the LV, since the OM artery (2 mm artery) runs along this margin, starting from the left mitral annulus and directed toward the apex. In addition, since the OM or major branches of it can pass the obtuse margin of the left ventricle and terminate at the inferior wall, to insonify these branches starting from a 4-chamber view, the probe should be angled inferiorly in order to tangentially transect the inferior wall of the left ventricle [18].

### 2.3. E-Doppler TTE versus Angiography: Pulsed-Wave Doppler Analysis

Color-guided PW Doppler was recorded in each of the 3 LAD segments: in each segment the maximal velocity was first recorded on the guidance of the localized aliasing of the color Doppler; in case of absence of the aliasing we recorded flow velocity in different point and chose the highest (in general, in the absence of aliasing the velocity is pretty uniform); then, the reference velocity was recorded immediately before or after the maximal and where the color flow appear normal.

The peak velocity of the diastolic waves was measured at these two sites; the percentage difference between them was calculated (diastolic peak at the first site − diastolic peak at the second site)/diastolic peak at the second site × 100 [4]. A percentage higher than 21% is considered abnormal and is called an ‘acceleration of the stenotic flow’ (AsF). The variability and inter- and intra-oberver reproducibility between these two measurements has been previously reported [1,3]. A percent increase of >108% (bootstrapped 95% CI > 87.5 to >108.82) at the first site was considered a critical stenosis, as previously validated vs. coronary angiography [19]. A percent increase of >21% (bootstrapped 95% CI > 20.0 to >22), but <108%, was considered a subcritical stenosis, as validated vs. intracoronary ultrasounds [3].

### 2.4. Feasibility Study Protocol 

The feasibility of the coronary blood flow assessment by E-Doppler TTE in LMCA, LAD, LCx and MO, before and after heart rate reduction, was evaluated using a three points score system, previously validated [20]. For color Doppler quality assessment, we attributed the scores as follows: score 1 = no signal, score 2 = sub-obtimal signal (width < 1 mm), score 3 = optimal signal (width > 1 mm) with no clutter artifacts. The maximal color Doppler flow signal length in each of the coronary segments studied was measured using calipers.

For PW Doppler recordings, the score we used was the following: score 1 = no signal, score 2 = spectral signal obtained but with scarce delineation of velocity curves and score 3 = optimal delineation of diastolic flow velocity curves at least.

All the color and PW Doppler quality assessments were performed by two different experienced operators (CC and PP).

#### 2.4.1. Study Protocol

All the patients enrolled had a heart rate of >65 bpm. On the same day, they underwent two complete examinations of the coronaries, namely, LMCA, LAD, LCx and OM, one before and the other after HR lowering. After excluding major contraindications to beta-blockers (in particular active asthma), they were premedicated with 100 mg PO of metoprolol and 0.7 mg of oral delorazepam. Thirty minutes later, the heart rate was checked again and, if it was <60 bpm, they were re-examined with a second complete echo-Doppler of the coronaries; contrarily, if the HR was still >60 bpm, intravenous 5 mg metoprolol was administered over 5 min and the second coronary E-Doppler TTE was executed after 20 min [3].

#### 2.4.2. Angiographic Analysis

All angiographic studies were performed the day after the E-Doppler TTE evaluation and interpreted in a blind manner, since they were performed as routine studies.

Coronary angiography was performed using the radial approach, and the coronary stenosis was visually assessed on the basis of multiple projections by one investigator who was unaware of the TTE Doppler results. Calipers were used in the case of doubt. The proximal LAD portion was angiocardiographically defined as extending from the left main coronary artery bifurcation up to and including the origin of the first major diagonal branch; the mid extending from the first diagonal to the last diagonal; and the distal portion as the remaining part of the vessel extending from the last major diagonal branch to the apex.

In the statistical analysis, continuous variables are expressed as mean values ± 1SD or median values (with non-parametric 95%CI or interquartile range [IQR]) as appropriate [21]. Correlations between scores were tested using the Spearman *rho* coefficient. Since both the pre-HRL scores and color flow length were not normally distributed, a Wilcoxon signed rank test was used to determine whether there was a difference between the scores and color Doppler length under two different conditions (before and after HRL) (a *p* value of <0.05 was considered significant).

The differences in the score value in the 3 LAD segments were assessed using Friedman’s test (which takes into account subject-specific correlations in non-normal distributions), as multiple segments per patient were used. If significant differences were found, cross-comparisons were made using Wilcoxon’s test.

The relationship between E-Doppler TTE and coronary angiography in evaluating coronary stenoses of the LAD was investigated using the Spearman *rho* coefficient.

Statistical calculations were performed using IBM SPSS statistics version 23, Armonk, NY: IBM Corp and MedCalc Statistical Software version 19.1.3 (MedCalc Software bv, Ostend, Belgium; 2019).

## 3. Results

The demographic, clinical and echocardiographic data are summarized in Table 1.

Beta-blockers significantly decreased the mean HR from 76 ± 5 to 57 ± 6 bpm (*p* < 0.001) (Table 2), without provoking any adverse side effects.

In our patients, HR lowering significantly and considerably increased color and pulsed wave Doppler signal intensity and quality in coronaries (Table 2).

Figure 1, Figure 2 and Figure 3 show representative color and pulsed wave Doppler images (baseline and enhanced studies by HR lowering).

Most importantly, HR lowering allowed a more complete exploration of each coronary vessel. The color Doppler flow length of the each explored vessel greatly increased after HR lowering, also greatly facilitating the PW Doppler positioning. In fact, color Doppler signal length increased from baseline to after HRL in the LAD (global vessel), from 13 mm (25) (median [IQR]) to 79 (17) mm (z = −4.46, *p* < 0.001), in the proximal LCx from 0 to 23 (13.5) mm (z = −4.37, *p* < 0.001) and in the obtuse marginal from 0 to 25 (12) mm (z = −4.17, *p* < 0.001) (Figure 4). It must be noted that color Doppler flow recording was extremely poor at the baseline both in the LCx (no flow at all) and in the OM (no flow, except in three patients) (Figure 4).

Quality of blood flow Doppler recording in the proximal and mid-LAD segments compared to the distal segment at baseline. The results of the Friedman test indicated that there was a statistically significant difference in PW quality scores across the three LAD segments at baseline (χ^2^ = 8.8, *p* = 0.01). Inspection of the median values showed an increase in the score (better quality) in the distal LAD with respect to the other, more proximally located LAD segments, confirmed with a post-hoc analysis (Figure 5). The color Doppler quality pre-HR reduction substantially paralleled these results (the quality of the color flow in the distal LAD at the baseline was better than in the other 2 LAD segments), but statistically showed only a trend (*p* = 0.06). Contrarily, after HR lowering, the quality score in the three LAD segments was similarly good (*p* = ns) for both the color (median score 3.0, 3.0 and 3.0, *p* = ns) and PW Doppler (median score = 3.0, 3.0 and 3.0, *p* = ns). This explains why the color Doppler score change from pre- to post-HR reduction in the proximal (median score change = 2, 95%CI 2 to 2) and mid-LAD (median score change = 2, 95%CI 2 to 2) was significantly larger than in the distal LAD (median score change = 1, 95% CI 0.54 to 2) (*p* = 0.0026 and *p* = 0.012, respectively); the PW Doppler paralleled the color Doppler results.

Comparison with coronary angiography: Improved quality of color Doppler signal was helpful in detecting coronary stenoses in the LAD. We verified this hypothesis in ten patients of the whole study group who underwent diagnostic coronary angiography (Table 3).

In these patients, the improved quality and length of the Doppler recording of blood flow velocity in the LAD after HR lowering was pivotal in detecting segmental acceleration at the stenosis site (transtenotic velocity) (Table 3).

Before enhancement, no adequate coronary blood flow by Doppler signal was recorded (no signal at all in four patients and poor signal with very limited length of the explored LAD color flow in the remaining six, hampering any detection of AsF) (Table 3). After HRL the coronary blood flow was detected satisfactorily, allowing AsF recording in five of the 10 patients (Table 3) (AsF identified, in three patients, critical stenosis, one located in the proximal and two in the mid; in two patients, mild stenosis in the proximal segment); in five patients, no AsF was detected (Table 3). The AsF perfectly predicted, both in terms of severity and location, the stenosis spotted at coronary angiography (Spearman correlation coefficient = 1, *p* < 0.01).

## 4. Discussion

For the first time, we have demonstrated that a reduction in HR (below 60 b/m) with appropriately oriented tomographic planes for visualizing coronaries and proper Doppler technology and setting enormously increases the feasibility in recording blood flow in coronaries by means of E-Doppler TTE. In particular, with this approach, LMCA and LAD blood flow can be obtained in all patients, with a huge clinical impact: increasing feasibility in assessing epicardial athero narrowing lesions and, if necessary, also expanding feasibility in assessing distal CFR (coronary flow reserve). In the LCx main trunk and terminal branches, the results of HR lowering were striking: blood flow in these arteries was visualized only with low HR. Thorough exploration of the LCx is more cumbersome than that of the LAD; nonetheless, the feasibility in LCx is considerably enhanced with a low HR, revealing very promising possible clinical applications.

### 4.1. Coronary Ultrasound Feasibility

Coronary blood flow Doppler recording by Doppler ultrasound has been considered a challenging task. However, different strategies over the years have considerably increased the feasibility of this approach. Transesophageal Doppler echocardiography was the first way to record flow in coronaries with [13,20] and without contrast [22]. However, transesophageal echo is invasive and visualizes only the proximal segments of the coronaries. Then, with the refinement of the Doppler module and probe [6], the transthoracic Doppler approach also became potentially useful for coronary blood flow Doppler recording [9], especially with the aid of contrast [13,20] and second harmonic Doppler technology [1,17,23]. However, contrast has its limitations: it is expensive, not always available on the market for business strategies (Levovist^®^ was withdrawn from the market several years ago), double IV lines for infusion are needed and, finally, it works poorly for the proximal part of the LAD, since the contrast-filled pulmonary infundibulum strongly attenuates the backscatter from the LMCA and proximal LAD, as reported in [4].

Therefore, notwithstanding the above-described refinement of the method, the ultrasound approach for coronaries has remained suboptimal and the examination challenging, limiting its clinical widespread applicability [5].

Finally, we noticed that a much better Doppler recording of coronaries could be obtained in patients with a low heart rate, such a strategy being adopted in addition to a good Doppler signal analyzer and appropriate tomographic planes [3]. In fact, the reduction in heart rate implies a lengthening of HR. In particular, the diastolic time can be calculated as the cardiac cycle (RR) minus the electromechanical systole (QS2). It has a curvilinear relationship with the heart rate (HR), increasing rapidly as the rate falls below 60 beats/min [10]. Additionally, this disproportionate increase in diastolic duration when the heart rate is below 60 b/m is of great aid in successful coronary insonification and blood flow Doppler recording. In fact, the coronaries normally move in and out of the tomographic plane as they follow the heart movements related to the contraction. Therefore, the coronary insonification during the diastole is limited. When the HR is high, the diastole is short, and the insonification time during such a brief diastolic period is of limited duration. This short scanning time hampers the proper distinction of the coronary flow signal, since the coronary signal is weak in intensity and also shows a small Doppler shift (low velocity); both conditions combine to create difficulty in distinguishing that signal from the background noise and clutter artifacts. The disturbing cluttering and wall thumps are enhanced by the strong movement of the cardiac wall, which is specifically more evident during an elevated HR, when the sympathetic drive has the tendency to rise, creating a more forceful contraction and more momentum of the cardiac wall’s movements. Therefore, a reduction in HR below 60/m enormously increases the signal noise ratio of the coronary Doppler signal not only by increasing the intensity, but also by reducing the background noise [3].

However, the hampering effect of the coronary motion on Doppler flow recording is not homogeneous. Studies of the mechanics of the heart have identified more motion at the base than at the apex [11,12]. This topographic effect is consistent with the observation that, among the left coronary vessels, the circumflex artery—which also follows the atrio-ventricular groove—has larger displacements and velocities [12]. We found this characteristic of LCx, the main problem with this vessel in terms of blood flow Doppler recording, which is also compounded by the complexity and variation of its course [18]. Therefore, it is mandatory to ensure a very low HR for adequate blood flow Doppler recording in this vessel.

Even though the OM runs from the base down to the apex, the portion generally insonified is the more proximal part, which has a greater caliber and is closer to the base of the heart (Figure 3). Therefore, the HR reduction is also extremely critical for this artery.

We have demonstrated, for the first time, that the feasibility of recording flow is higher for the distal than the proximal and mid-LAD segments at the baseline condition (before HR lowering) (Figure 5). Higher mobility of the more proximal segments drives this effect, while a significant more distal portion of the LAD runs along the anterior surface of the left ventricular wall and is not subject to these sudden motions, and consequently exhibits lower average velocities [12]. This explains why, in the past, blood flow Doppler recording in the distal LAD segments was carried out with a certain feasibility, even without advanced enhancing strategies [24,25,26]. Since the quality of blood flow Doppler recording in the three LAD segments after HR lowering is highly and uniformly improved, the reduction in HR below 60 b/m for improving blood flow Doppler recording is much more beneficial for the proximal and mid-LAD segments than for the distal one.

Clinical impact of coronary Doppler signal enhancement by lowering heart rate. We have shown, for the first time, the practical effect of Doppler enhancement in expanding the diagnostic information in the clinical arena. In fact, such clinical gain was verified in the same patients when the enhancing effect was compared with the pre-enhancement status. One limitation of feasibility studies is that the real clinical gain of the enhancer is verified in a population that is different from that one in which the original enhancing effect was shown [3]. Therefore, the control status (pre-enhancement status) is no longer present in this context. This creates some perplexity about the real clinical utility of the enhancer (low HR) and a tendency to consider other spurious factors as the cause of the good quality apart from the enhancer (good ultrasound window, etc.). In this study, thanks to HR lowering in patients with no flow or minimal suboptimal flow detected at baseline, we were able to record flow to the extent that different grades of AsF (from mild to severe) were also recorded and quantified. These perfectly predicated, in terms of the severity and location, the epicardial atherosclerosis involvement as assessed by angiography (Table 3) [2,3,4,27]. Therefore, the coronary flow Doppler recording is improved to the extent that abnormality of flow can be recorded well enough to properly predict the angiographic coronary abnormality. In this respect, HRL is pivotal, since it is particularly effective in those segments (proximal and mid) more frequently affected by athero [15] (Table 3).

Tomographic plane orientation is another big issue which need to be resolved in order to optimize feasibility. Preliminary reports have documented the refinement of the tomographic plane orientation technique in visualizing the entire LAD along with the LMCA [3]. The apical approaches and parasternal approach that exploit the left lung cardiac notch represent a huge step forward in LMCA (Figure 2) and LAD visualization by ultrasound [3] (Figure 1). Conversely, the visualization of the proximal–mid circumflex and especially of the obtuse marginal has been poorly standardized and only anecdotally reported. We used a very high parasternal approach for the proximal LCx in order to properly insonify the left atrial–ventricular sulcus where the artery runs [28]. The recognition of the left atrial appendix is a good internal marker to guide the insonification [28] (Figure 2).

### 4.2. Previous Studies

This is the first study that has used a reduction in HR below 60/m in order to increase the feasibility of blood flow Doppler recording in coronaries; the only other studies that have obtained a good enhancement with no HRL have been those that used contrast enhancement, especially along with second Harmonic filtering technologies [1,17,23,29]. However, these contrast techniques are expensive and time-consuming. In other studies, the enhancement has only been attempted through new tomographic planes, but with a minor impact [30]. However, before HR lowering, Doppler recording of coronary blood flow remains challenging.

### 4.3. Clinical Implications

This study has important clinical implications.

The LMCA and LAD can be explored with a feasibility of 100%, with HR lowering on top of convergent color Doppler and improved tomographic planes. This offers a very important clinical insight: first, direct athero evaluation (from mild to severe stenoses) is rapidly obtained by means of AsF detection [3,4]. More importantly, the diffuse athero, usually unseen even by invasive coronary angiography [31], can be reliably detected by adding the assessment of the distal CFR: a blunted CFR with even mild stenosis as assessed by AsF is an expression of diffuse atherosclerotic involvement, as recently preliminarily demonstrated [32].

The more successful recording in the proximal–mid LCX main trunk is a step forward in evaluating the diffusion of atherosclerosis in coronaries. AsF detected in this part of the artery can have an important diagnostic and prognostic implication in distinguishing ischemic from non-ischemic dilated cardiomyopathy, along with LAD blood flow evaluation. However, validation studies are needed.

Clinically, the obtuse marginal blood flow Doppler recording is particularly appealing, since it can allow the assessment of the distal CFR in the LCx, as preliminarily reported [16,33,34,35]. However, the high HR and increased respiratory rate during adenosine make it extremely difficult to record the hyperemic flow in this artery. In a previous experience, without a protocol of HR lowering, the feasibility of CFR was around 70% in this artery [33,35], and this is in agreement with our own preliminary experience [36], but without HR lowering being systematically obtained.

### 4.4. Limitations

The main limitation of this study is that a HR of < 60 b/m is not always easily obtainable. Premedication is necessary in most patients. Additionally, that is time-consuming. Beta-blockers are the first strategy and that can have a rapid effect; but beta-blockers cannot work properly or are contraindicated (e.g., by hypotension or asthma) in some cases; so the use of Ivabradine 7.5 mg two times a day for at least 3 days before the examination, with the last dose on the morning of the examination, is, in our view, the best alternative strategy [37]. On the day of the exam, if HR is still suboptimal, beta-blockers in the absence of hypotension and other beta-blocker contraindications can be added on top of ivabradine (25 mg of metoprolol every 15 min until the HR falls below 60/m or blood pressure tends to fall below 100 mmHg systolic). Finally, any drug possibly accelerating the HR (thyroid hormones, vasodilators such as dihydropiridine calcium blockers, alpha blockers, certain antibiotics such as azithromycin, etc.) must be suspended at least 3 days before the examination if this is possible. If the patient shows an unexplained elevated HR, recreational drugs should eventually be investigated in the history or even dosed in urine (cocaine). In the emergency room, unless the patient is sedated and has been successfully treated with beta-blockers, E-Doppler TTE with HRL has limited utility, since the patients have a HR that is usually above 60 b/m. Atrial fibrillation with high HR is a contraindication for blood flow Doppler recording and AsF detection unless the patient is appropriately treated in order to increase the refractory period of the atria-ventricular node and so prolong the diastolic time by at least some beats. In some patients, however, coronary blood flow can also be unexpectedly detected with a HR above 60 b/m.

The effect of low HR on BF Doppler recording feasibility in the right coronary artery, particularly in the posterior descending coronary artery, has not been tested in this study. Further studies are needed.

The HRL is neutralized during the adenosine step of the CFR assessment that creates disturbing tachycardia compounded by more pulmonary interference (more breathing drive). This is particularly true for very mobile and less well-visualized arteries (lung tissue interference), such as the OM and right coronary artery. In these cases, apart from HRL, the injection of contrast during the adenosine infusion could be tried. Further studies are needed.

The sample size of this study is low and further larger confirmatory studies are warranted.

Although, in this limited series, there were no collateral effects worth mentioning after drug administration, beta-blockers—especially at high dosage—can, in rare cases, provoke hypotension (always asymptomatic), so repeated measurement of blood pressure is warranted during the examination. In case hypotension takes place and adenosine infusion is planned, a blood expander such as polygeline (300–500 cc) has to be IV infused. Wheezing is never a problem after beta-blockers administration in order to reduce HR, since we carefully clinically evaluate the lung status and exclude those patients with signs of expiratory obstruction (even if minimal).

## 5. Conclusions

Drug-induced heart rate lowering greatly enhances transthoracic blood flow Doppler recording in coronaries, making E-Doppler TTE suitable for accelerated stenotic flow and flow reserve assessment not only in the left anterior descending, but also in the circumflex branch.

## Figures and Tables

**Figure 1 diagnostics-13-00670-f001:**
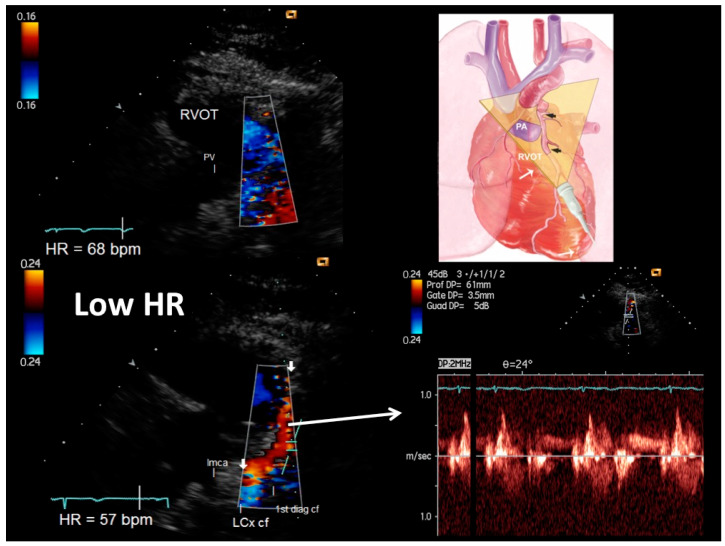
An example of proximal–mid-LAD color flow (plane orientation at the top right corner) at higher HR (68 b/m) (upper right) and after HR reduction (57 b/m) (bottom right): at the baseline with higher HR no flow is detected; after HR lowering (<60 b/m), optimal color signal is recorded in the LAD so allowing color-guided pulsed-wave Doppler sampling of optimal quality (bottom right). Additionally, the first thin diagonal branch is depicted along with the origin of the left circumflex coronary artery whose flows are coded in blue since are away from the probe. The left main coronary artery (directed posteriorly) is evident in B-mode. LMCA = left main coronary artery; LCx cf = proximal left circumflex coronary artery color flow; 1st diagonal cf = first diagonal color flow; LMCA = left main coronary artery; RVOT = right ventricular outflow tract; pv = pulmonary valve.

**Figure 2 diagnostics-13-00670-f002:**
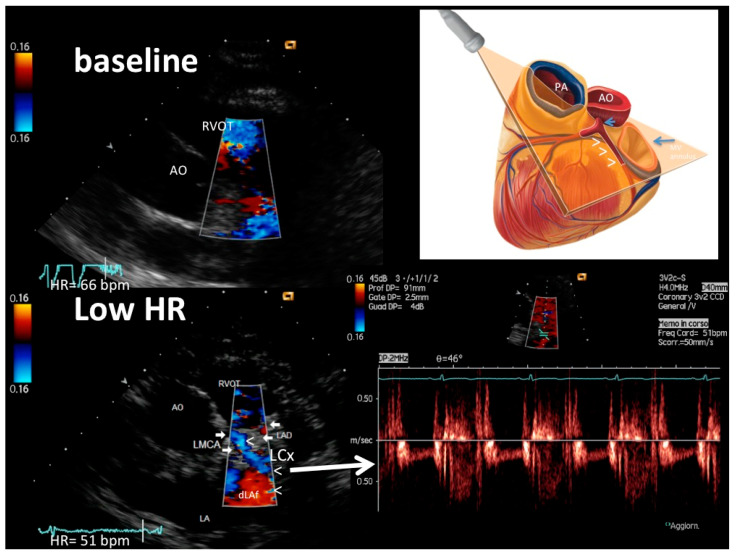
An example of LMCA, proximal–mid LCx and proximal LAD color flow (plane orientation at the top right corner) at higher HR (66 b/m) (upper left) and after HR reduction (51 b/m) (bottom right): at the baseline with higher HR, no coronary flow is detected; after HR lowering (<60 b/m), optimal color signal is recorded in the LMCA (arrows), LCx (arrowheads) and very proximal LAD (arrows, signal in red); thanks to good color guidance optimal pulsed-wave Doppler quality in LCX is obtained (bottom right). The color signal is coded in blue both in LMCA and LCx and the LCx PW tracing is below the zero line because flow is away from the transducer. In the schematic cartoon (top right), the arrowheads indicate the LCx, the long arrow the mitral annulus and the short arrow the LMCA. LMCA = left main coronary artery color flow; LAD = proximal LAD color flow; LCx = proximal–mid left circumflex coronary artery color flow; RVOT = right ventricular outflow tract; pv = pulmonary valve; Ao = aorta; LA = left atrium.

**Figure 3 diagnostics-13-00670-f003:**
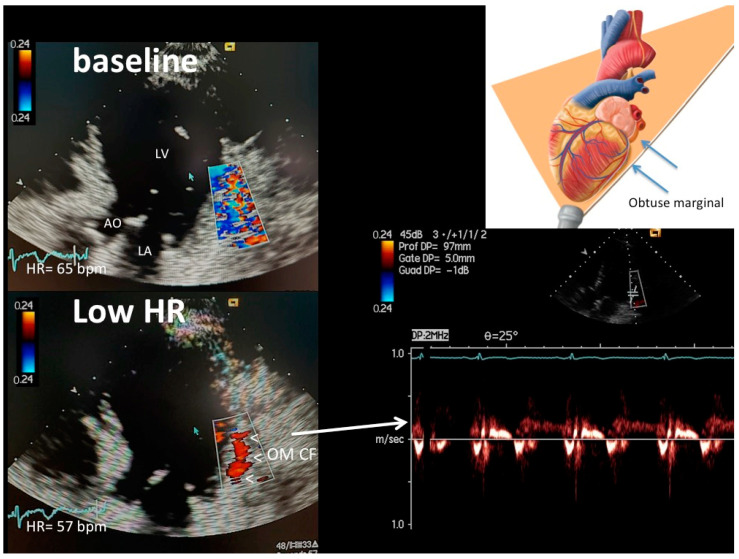
An example of OM color flow (plane orientation at the top right corner) at high HR (65 b/m) (upper left) and after HR reduction (57 b/m) (bottom left): at the baseline with higher HR, no flow is detected (only clutter artifacts); after HR lowering (<60 b/m), optimal color signal is recorded (indicated by arrowheads) in the proximal OM so allowing color-guided pulsed-wave Doppler sampling of optimal quality (bottom right). OM CF = color flow in the obtuse marginal; LV = left ventricle; AO = aorta; LA = left atrium.

**Figure 4 diagnostics-13-00670-f004:**
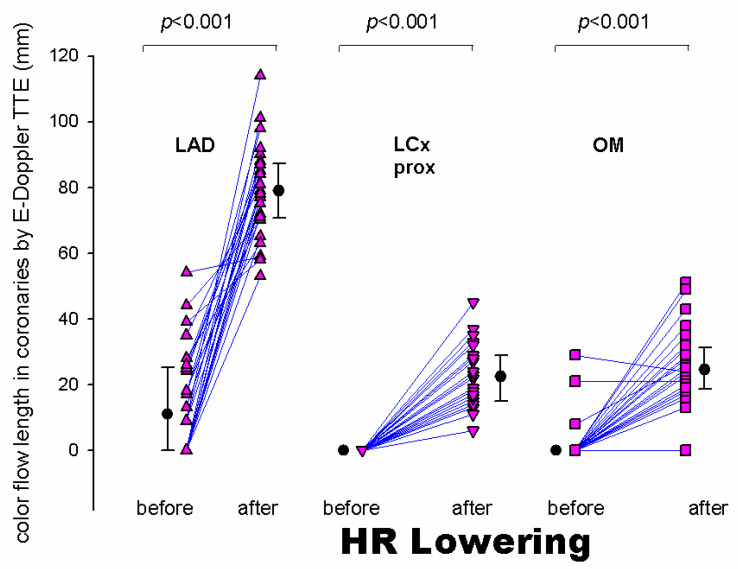
Individual value bar graph (and median [IQR]) of the color flow length measured in the whole LAD (triangles vertex up), proximal LCx (triangles vertex down) and the Obtuse Marginal (squares) before and after HR lowering. The length of the artery explored by E-Doppler TTE in terms of color flow is enormously and significantly expanded after HR lowering. LAD = left anterior descending coronary artery; LCx = left circumflex coronary artery; prox = proximal; OM = obtuse marginal branch of the LCx; HR = heart rate; IQR = interquartile range.

**Figure 5 diagnostics-13-00670-f005:**
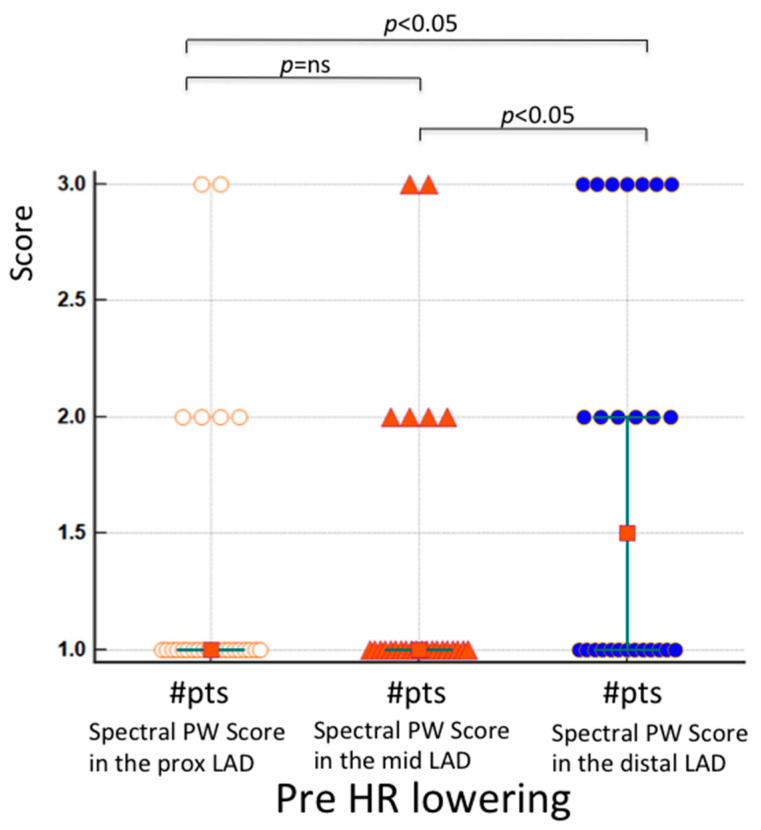
Individual value bar graph showing color-guided PW Doppler quality score pre-HR lowering in the 3 segments of the LAD in 26 patients. The mean rank (1.82 and 1.82) and the median score (1 for both) are similarly low in the two first category (proximal and mid-LAD), while these are significantly higher in the distal LAD (mean rank 2.32 and median 1.5, 95 CI 1–1.5). # = number of patients; white circles= score of proximal LAD segments; orange triangles= score of mid LAD segments; blue circles= score of distal LAD segments.

**Table 1 diagnostics-13-00670-t001:** Demographic, clinical and echocardiographic data in 26 patients.

Age, years (years)	62 ± 14
Gender	
Males, #patients (%)	17 (65%)
Females, #patients (%)	9 (35%)
BMI	26 ± 4
Diabetes, #patients (%)	4 (15%)
Hypertension, #patients (%)	18 (69%)
Typical Angina, #patients (%)	3 (12%)
Atypical Angina, #patients (%)	6 (24%)
Non-Anginal chest pain, #patients (%)	1 (4%)
Hystory of CAD, #patients (%)	18 (69%)
Previous PTCA, #patients (%)	2 (8%)
Previous myocardial infarction, #patients (%)	7 (27%)
Blood work	
Glycemia (mg/dL)	110 ± 43
Total cholesterol (mg/dL)	190 ± 45
HDL cholesterol (mg/dL)	46 ± 14
LDL cholesterol (mg/dL)	116 ± 46
Triglycerides (mg/dL)	141 ± 80
Echocardiographic data	
LVEDd (mm)	51 ± 5
LVESd (mm)	33 ± 6
LVEF, %	58 ± 12

BMI = body mass index; CAD = coronary artery disease; PTCA = percutaneous coronary angioplasty; LVEDd = end diastolic left ventricular diameter; LVESd = end systolic left ventricular diameter; LVEF = left ventricular ejection fraction. # = number of patients.

**Table 2 diagnostics-13-00670-t002:** Color Doppler quality in LMCA, LAD and LCx, before and after heart rate lowering.

	Color Doppler			PW Doppler			CF Length
	Score 1	Score 2	Score 3	Score 1	Score 2	Score 3	mm
	# (%)	# (%)	# (%)	# (%)	# (%)	# (%)	
LMC b, 26 seg	24 (92)	1 (4)	1 (4)	24 (92)	1 (4)	1 (4)	-
LMC a, 26 seg	0 (0)	12 (46)	14 (54) *	0 (0)	15 (58)	10 (38) *	-
LAD prx, b 26 seg	20 (77)	6 (23)	0 (0)	20 (77)	4 (15)	2 (8)	-
LAD prx, a 26 seg	0 (0)	0 (0)	26 (100) *	0 (0)	2 (8)	24 (92) *	-
LAD mid, b 26 seg	21(81)	3(11)	2 (8)	22 (85)	4(15)	0	
LAD mid, a 26 seg	0 (0)	3 (11)	23 (88) *	0	2 (8)	24 (92) *	-
LAD dst, b 26 seg	14 (54)	9 (35)	3 (11)	13 (50)	6 (23)	7 (27)	-
LAD dst, a 26 seg	1 (4)	1 (4)	24 (92) *	1 (4)	2 (8)	23 (88) *	-
LCx b, 26 seg	26 (100)	0	0	26 (100)	0	0	0
LCx a, 26 seg	0	9 (35)	17 (65) *	0	6 (23)	20 (77) *	23 (9.07) **
OM b, 26 seg	23 (88)	3 (11)	0	24 (92)	1 (4)	1 (4)	0
OM a, 26 seg	1 (4)	12 (46)	13 (50) *	1 (4)	15 (58)	10 (38) *	25 (12) **

b = before HR lowering; a = after HR lowering; seg = coronary segments; # = number of coronary segments; (%) = percentage; CF = color flow; LMC = left main coronary artery, LAD = left anterior descending subdivided into proximal, mid and distal segments; LCx = proximal circumflex; OM = obtuse marginal; * = *p* < 0.001 versus score before HR lowering; ** = *p* < 0.001 versus color flow length before HR lowering; color length is expressed as median (IQR).

**Table 3 diagnostics-13-00670-t003:** List of angiographic and E-Doppler TTE LAD findings in 10 patients.

Coronary Angiography				E-Doppler TTE					
# of Patients	Prox LAD (% st)	Mid-LAD (% st)	Distal LAD (% st)	Before HR Lowering		After HR Lowering			
				LAD CF Length (mm)	AsF in the LAD (%)	LAD CF Length (mm)	AsF Prox LAD (%)	Asf Mid-LAD (%)	AsF Dis LAD (%)
1	0	50	0	13.00	0	79.00	104.55	34.38	0.00
2	0	0	0	0.00	0	79.00	0.00	0.00	0.00
3	0	0	0	44.00	0	70.00	0.00	0.00	0.00
4	0	0	0	0.00	0	101.00	0.00	0.00	0.00
5	50	0	0	0.00	0	77.00	168.1	0.00	0.00
6	0	0	0	35.00	0	81.00	0.00	0.00	0.00
7	40	0	0	9.00	0	92.00	37.14	0.00	0.00
8	70	75	0	9.00	0	114.00	164.00	83.33	7.32
9	80	0	70	0.00	0	72.00	971.43	0.00	0.00
10	0	0	20	18.00	0	71.00	0.00	0.00	0.00

# of patients = sequential number of patients; % st = percentage of luminal diameter narrowing visually estimated; Prox = proximal; LAD = left anterior descending coronary artery; AsF = acceleration of stenotic flow; Dis = distal, CF = color flow; B = before; A = after; HR = heart rate.

## Data Availability

Not applicable.

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
