# Peer review of "Heart Rate Lowering Significantly Increases Feasibility in Doppler Recording Blood Flow Velocity in Coronaries during Transthoracic Doppler Echocardiography"

_diagnostics, 2023, doi:10.3390/diagnostics13040670_

Round 1

Reviewer 1 Report

See attached file

Author Response

Title: Heart Rate Lowering Significantly Increases Feasibility in Doppler Recording Blood Flow Velocity in Coronaries During Transthoracic Doppler Echocardiography

Authors: Carlo Caiati et. al.

Journal: diagnostics

This paper is an interesting piece of work and overall well-written. The following comments refer to the statistical analysis of the paper.

Statistical Review

1) The sample size the results are based, is quite low and this fact should be added to the study limitations

1) Authors’ answer: Tanks for the comment; we agree; that has been added in the limitations of the discussion (page 15, in red).

2) In Table 2, there are cells with small or zero frequencies. Has a continuity correction performed in the chi-square or other tests to tackle this issue?

2)Authors’ answer: Thanks very much; we have fixed the problem by reanalyzing all the paired data of Table 2 by using the Wilcoxon signed rank test. That has been also specified in the statistical subsection of the methods (page 5, in red).

3) If we base our analysis on the 156 coronary segments, by setting a database with 156 rows, a column for segment identification with 6 levels, a column for HR status with 2 levels (before / after), a column for score with 3 values (1,2,3) and possibly other columns incorporating demographic and clinical information presented in Table 1, then using an ordinal model with score as the response and covariates appropriately selected, the authors could make comparisons with better statistical properties from the ones presented

3) Authors’ answer: Thank you: the suggestion is excellent and interesting, and we will keep it in mind for other researches. However, in this case the available data are paired ("before" and "after", in the same segments of the same patients), and creating the database as defined in your valuable suggestion would force us to also insert an ID column (first level descriptor) on which to base all the comparisons, including those between the data of the 6 different segments (which, in turn, are paired in the same first-level unit). In this way, a "nested and paired" multilevel ordinal model would be defined, i.e. with the data segments nested in pre- and post-HR reduction patient data. The problem is that multilevel models (on an ordinal basis) do not yet support paired data, and in any case even models for independent 2nd level units (simpler as conception) require a larger size than the available sample (see: D. Hedeker, "Handbook of multilevel analysis" Springer, 2008).

4) It is not clear the message conveyed in Figure 5

4) Authors’ answer: we have replaced that Figure with a more understandable graph that eliminates the box-whisker plots, that can be confusing.

5) Since the results of E-Doppler TTE and Angiography coincide, Table 4 is not needed. Authors could simply report the distribution of location and severity of LAD stenosis in the 10 patients and just mention that the classification of the two methods are identical.

5)Authors’ answer: Thanks for the suggestion; we agree with the reviewer and accordingly we have modified the text: the table 4 has been taken out and we have reported in the text the stenosis characteristics (that can be found also in the in table3) and we have added that the prediction by ASF is perfect (page 12 , in red) .

Reviewer 2 Report

Thank you for the opportunity to review this manuscript.

The authors are to be commended for demonstrating how the feasibility of echocardiographic assessment of the coronary arteries can be increased by lowering heart rate. This is important work in the dissemination of non-invasive imaging of the left coronary arteries without the administration of contrast agents.

General:

The authors draw on a wealth of knowledge and experience, which makes the text somewhat complex. The main aim should be to encourage cardiologists to use this method, but the actual presentation of the results could be confusing for readers who are not familiar with the field. The manuscript text could be made more concise, especially the discussion section. Furthermore, there are inconsistencies in the use and explanation of abbreviations (e.g. line 259: CFR not explained, or BF).

In addition, there are some important considerations that should be addressed as follows:

a)Abstract:

The following conclusion "... may have wider clinical application" is somewhat over-enthusiastic given the limited data, but further studies with larger samples are needed to confirm these results.

b) Methods:

1. Line 68/69: The concept of AsF should be explained in more detail.

2. Line 191: Why was delorazepam administered? Please clarify.

3. L168 to 173: The method and type of bootstrapping should be clarified, because you would get the impression that bootstrapping was carried out on two measurements.

4. Although the rationale for angiographic analysis is understandable, the authors have already demonstrated the impact of beta-blocker use on the detection of coronary stenosis with echocardiography in reference #3. Furthermore, was the angiography performed at the same time as the echocardiography or delayed? This can have different implications.

5. L208: why median +- standard deviation? Inter-quartile range would be more appropriate.

6. L208 and Table4: Chi-square test would not appropriate due to small number (<5) per cell.

7. L215-216: Why T-test? Was the length normally distributed? Please clarify.

d) Results:

1. There is no mention of any potential adverse effects of using beta blockers to lower heart rate in the study population.

2. L 305 to 322: The results should be presented more concisely, speculations in line 316 should be made in the Discussion section.

3. Figure 5: Boxplots are not particularly informative as this score only has a limited range of 1 to 3 and is presented as integers/whole numbers.

e) Discussion:

1. What does Reference #21 add to the actual manuscript? There is no further description of the method or results.

2. The discussion section includes a lengthy review of the history and current state of coronary blood flow Doppler recording.

Author Response

Thank you for the opportunity to review this manuscript.

The authors are to be commended for demonstrating how the feasibility of echocardiographic assessment of the coronary arteries can be increased by lowering heart rate. This is important work in the dissemination of non-invasive imaging of the left coronary arteries without the administration of contrast agents.

General:

#1 The authors draw on a wealth of knowledge and experience, which makes the text somewhat complex. The main aim should be to encourage cardiologists to use this method, but the actual presentation of the results could be confusing for readers who are not familiar with the field. The manuscript text could be made more concise, especially the discussion section. Furthermore, there are inconsistencies in the use and explanation of abbreviations (e.g. line 259: CFR not explained, or BF).

Authors’ answer (#1):  Thanks for the suggestion; accordingly we have considerably shortened the Discussion section that now is decisively much more concise and the results section as well. In addition we have added legends for the abbreviations CFR and BF as suggested.

In addition, there are some important considerations that should be addressed as follows:

a)Abstract:

#2 The following conclusion "... may have wider clinical application" is somewhat over-enthusiastic given the limited data, but further studies with larger samples are needed to confirm these results.

 Authors’ answer (#2):  Thanks for the suggestion; we have added a sentence that expresses the need for further, larger confirmatory study. (page 1 in red)

b) Methods:

1. Line 68/69: The concept of AsF should be explained in more detail.

Authors’ answer (Methods #1): We thank very much the reviewer for this appropriate observations; accordingly we have implementd the Methods section with more appropriate description on how to obtain and calculate the AsF (Methods page 4, in red )

 2. Line 191: Why was delorazepam administered? Please clarify.

Authors’ answer (Methods #2): Thanks for asking. In some patients the anxiety component can be consistent and can contribute to maintain the HR above the 60b/m notwithstanding the beta blockers medications.

 3. L168 to 173: The method and type of bootstrapping should be clarified, because you would get the impression that bootstrapping was carried out on two measurements.

Authors’ answer (Methods #3): Thanks for asking. In that previous study the bootstrapping was used since the resulting estimated cutoff is known to be very susceptible to changes in the study population and it is recommended to use e.g. bootstrapping techniques to validate the results. So its recommended to provide 95% confidence interval for the estimated 'optimal' cutoff. To do the bootstrapping calculations we used at that time a bootstrapping replications of 5000 times and a random number seed of 968.

However in this study we did not perform any bootstrapping calculations but simply apply a cutoff obtained and validated in a previous study.

4. Although the rationale for angiographic analysis is understandable, the authors have already demonstrated the impact of beta-blocker use on the detection of coronary stenosis with echocardiography in reference #3. Furthermore, was the angiography performed at the same time as the echocardiography or delayed? This can have different implications.

Authors’ answer (Methods #4): Thanks for asking. Angiography was performed the day after the execution of E-Doppler TTE. That has been added in the methods section page 5, in red.

5. L208: why median +- standard deviation? Inter-quartile range would be more appropriate.

Authors’ answer (Methods #5): Thanks for this observation; accordingly we have used the median with the interquartile range (IQR) all over the manuscript. Also we used MedCalc for median's CI estimation, and under your suggestion we have now corrected the used methodology (see  Campbell and Gardner, 1988). This has been specified in the statistical subsection (page 5 in red).

6. L208 and Table4: Chi-square test would not appropriate due to small number (<5) per cell.

Authors’ answer (Methods #6): Thanks for noticing that. You are perfectly right. We have replaced that analysis with the non parametric spearman correlation. Accordingly we have modified the results section (page 12, in red) and we have described this analysis in the statistical subsection of the results (page 5, in red).

7. L215-216: Why T-test? Was the length normally distributed? Please clarify.

 Authors’ answer (Methods #7): Thanks for this comment; not all the length series were normally distributed; in particular the pre HRL were not; so we have reanalyzed all the data of the vessel length by Wilcoxon Signed Rank Test; the results however did not change. We have mentioned this in the statistical section (page 5, in red) and we added the z value of the test in the results section (page 9, in red).

d) Results:

1. There is no mention of any potential adverse effects of using beta blockers to lower heart rate in the study population.

Authors’ answer (Results #1): We have got large experience about beta blockers; their collateral effects are rare when transiently administered; the major collateral effects in our Lab is hypotension (systolic blood pressure <100 mmHg) especially when high dosages are titrated; this hypotension is always asymptomatic though. In these cases if adenosine infusion is planned for coronary flow reserve assessment, before adenosine infusion we usually treat hypotension with blood volume expansion (Polygeline infusion). This is generally successful but it takes time. Wheezing is never a problem since we carefully clinically evaluate lungs and exclude those patients with signs of expiratory obstruction (even minimal). However in the reported limited series there were no collateral effects worth mentioning.

We have added in the results section (page 6, in red): ”with no adverse side effects”. In addition more extensive comments on the potential side effects of beta blockers as used in our lab has been added in the limitations section (page 15, in red) .

2. L 305 to 322: The results should be presented more concisely, speculations in line 316 should be made in the Discussion section.

Authors’ answer (Results #2): Many thanks for this suggestion; we agree with the point of view of the reviewer; accordingly we have taken out all the comments regarding Table 2 and the speculations. The speculations as suggested have been added to the discussion (page 14, in red). So now the results are much more concise.

3. Figure 5: Boxplots are not particularly informative as this score only has a limited range of 1 to 3 and is presented as integers/whole numbers.

 Authors’ answer (Results #3): Thanks for this observation; we agree and accordingly we have eliminated that graph and replaced it with an individual value bar-graph with the median value (and 95 CI of the median). (Figure 5)

e) Discussion:

1. What does Reference #21 add to the actual manuscript? There is no further description of the method or results.

Authors’ answer (Discussion #1): We totally agree; that reference has been taken out.

2. The discussion section includes a lengthy review of the history and current state of coronary blood flow Doppler recording.

Authors’ answer (Discussion #2): Thanks for this comment. We totally agree with him. Accordingly we have consistently shortened that part of the discussion by eliminating 280 words (all the modifications page 12, in red).

Round 2

Reviewer 2 Report

Thank you for the opportunity to review this manuscript. The authors have responded adequately to our comments.